# Nanostructured Iron Sulfide/N, S Dual-Doped Carbon Nanotube-Graphene Composites as Efficient Electrocatalysts for Oxygen Reduction Reaction

**DOI:** 10.3390/ma14092146

**Published:** 2021-04-23

**Authors:** Gyu Sik Chae, Duck Hyun Youn, Jae Sung Lee

**Affiliations:** 1Division of Environmental Science and Engineering, Pohang University of Science and Technology (POSTECH), Pohang 37673, Korea; she213@postech.ac.kr; 2Interdisciplinary Program in Advanced Functional Materials and Devices Development, Department of Chemical Engineering, Kangwon National University, Chuncheon, Gangwon-do 24341, Korea; 3School of Energy & Chemical Engineering, Ulsan National University of Science and Technology (UNIST), Ulsan 44919, Korea

**Keywords:** fuel cells, oxygen reduction reaction, iron sulfide, carbon nanotube–graphene composites, N, S dual doping

## Abstract

Nanostructured FeS dispersed onto N, S dual-doped carbon nanotube–graphene composite support (FeS/N,S:CNT–GR) was prepared by a simple synthetic method. Annealing an ethanol slurry of Fe precursor, thiourea, carbon nanotube, and graphene oxide at 973 K under N_2_ atmosphere and subsequent acid treatment produced FeS nanoparticles distributed onto the N, S-doped carbon nanotube–graphene support. The synthesized FeS/N,S:CNT–GR catalyst exhibited significantly enhanced electrochemical performance in the oxygen reduction reaction (ORR) compared with bare FeS, FeS/N,S:GR, and FeS/N,S:CNT with a small half-wave potential (0.827 V) in an alkaline electrolyte. The improved ORR performance, comparable to that of commercial Pt/C, could be attributed to synergy between the small FeS nanoparticles with a high activity and the N, S-doped carbon nanotube–graphene composite support providing high electrical conductivity, large surface area, and additional active sites.

## 1. Introduction

The oxygen reduction reaction (ORR) is crucial for electrochemical energy conversion and storage devices including fuel cells and lithium-air batteries [1,2]. The fuel cell is a promising energy conversion device due to its high energy density, rapid start-up, and zero emissions [3,4,5]. ORR typically occurs at the cathode of fuel cells heavily loaded with platinum [6,7], but its high cost, low abundance, and instability have made the fuel cell a highly expensive device [8]. Thus, it is essential to develop non-precious metal-based ORR catalysts offering high activity and stability for more rapid dissemination of fuel cells [9].

A variety of materials have been investigated as alternative non-Pt catalysts for ORR, including oxides [10], nitrides [11,12,13], sulfides [14,15], carbides [13,16] of transition metals, metal–nitrogen–carbon catalysts (MNC) [17,18,19], and metal-free catalysts [20,21]. Various transition metal sulfides (TMS) of Mo, Fe, Co, Ni, and V have been explored as ORR catalysts due to their earth-abundance, low cost, and considerable activity [14]. Nanostructured TMS have also been considered to further improve the ORR activity, due to increased number of active sites compared to their bulk counterparts. Another way to improve ORR activity is to combine TMS with carbon supports including carbon nanotube (CNT), graphene (GR), and amorphous carbon [15]. Carbon supports can provide high electrical conductivity and large surface area to disperse TMS, enhancing ORR activity [15,22,23]. They can also act as a growth mediator, reducing TMS particle aggregation. Heteroatom (N and S) doping into the carbon supports further increases active sites and hence ORR activity by modulating their electronic structure [24,25,26]. Therefore, fabricating nanostructured TMS on heteroatom-doped carbon supports could offer a good approach to improve the ORR activity of TMS.

This paper proposes a simple fabrication method for FeS nanoparticles loaded onto the N, S dual-doped CNT–GR support (FeS/N,S:CNT–GR). Nanostructured FeS/N,S:CNT–GR was obtained by simply annealing a mixture of Fe precursor, thiourea, and carbon supports (CNT and graphene oxide (GO)) at 973 K under flowing N_2_ and subsequent acid treatment for 30 min. The synthesized FeS/N,S:CNT–GR catalyst showed an impressive ORR activity and stability. It offers various merits in terms of synthesis and ORR performance: (1) The proposed fabrication method is simple and economical. FeS crystallization, reduction of GO to GR, and N, S dual-doping into the CNT–GR support were simultaneously achieved in one-pot reactions under N_2_ atmosphere. Unlike previous reports, no further chemicals for GO reduction and N, S dual-doping or toxic gas for FeS crystallization are required. (2) Our FeS/N,S:CNT–GR catalyst exhibits one of the best performances among the iron-based TMS catalysts, with small onset and half-wave potential values of 0.972 and 0.827 V, respectively, and a good stability for 6000 potential cycling. The high ORR activity of FeS/N,S:CNT–GR is probably due to synergy between active FeS nanoparticles and the N, S dual-doped CNT–GR support providing large surface area and high electrical conductivity of the catalyst. The N,S:CNT–GR by itself further stimulates ORR performance through the N, S dual doping, and mediates FeS growth to mitigate particle aggregation. Thus, considering the simple fabrication method and the high ORR performance, FeS/N,S:CNT–GR could be a promising non-precious metal electrocatalyst for ORR.

## 2. Materials and Methods

### 2.1. Synthesis of FeS/N,S:CNT–GR

Graphene oxide (GO) was synthesized by Hummer’s method [27] and commercial CNT (CMP-301F, Hanwha Nanotech, Incheon, Korea) was acid treated with 90% nitric acid and 99% sulfuric acid solution, 1:3 *v/v* ratio, at 393 K for 3 h to eliminate residual metal species before use [28]. Typically, GO (116 mg) and CNT (116 mg) were dispersed in 5 mL ethanol. FeCl_2_·4H_2_O (1 g) was dissolved in 5 mL ethanol and added to the CNT–GO solution. Thiourea (1.5 g) was added to the solution and the solution was stirred for 1 h. The resulting solution was dried in an oven at 373 K to evaporate ethanol and annealed at 973 K (at a heating rate of 10 °C min^−1^) for 3 h under N_2_ atmosphere. The sample was stirred in 0.5 M sulfuric acid solution for 30 min to obtain pure, crystalline phase FeS loaded onto the N,S:CNT–GR support. FeS/CNT and FeS/GR were prepared following an identical method except only CNT or GO was included during the synthesis. Bare FeS was synthesized identically without any carbon support. Nominal weight content of FeS in the supported FeS catalysts was fixed to 50%, and measured weight contents (by inductively coupled plasma optical emission spectroscopy, ICP) were 52, 50, and 54 wt% for FeS/N,S:CNT–GR, FeS/N,S:CNT, and FeS/N,S:GR, respectively.

### 2.2. Catalyst Characterization

Crystalline structures of the prepared catalysts were determined by X-ray diffraction (XRD, pw 3040/60 X’pert diffractometer, Malvern PANalytical, Malvern, UK) with Cu Kα radiation. Structural details were investigated by transmission electron microscopy (TEM, JEM-2100F, JEOL Ltd., Tokyo, Japan) and energy dispersive microscopy (EDS, Oxford Instruments, x-Max T-80, Abingdon, Oxon, UK). Surface chemical composition and electronic state were measured by X-ray photoelectron spectroscopy (XPS, K-alpha, Thermo Fisher Scientific, Waltham, MA, USA) and Raman spectroscopy (Alpha300R, WITec, Ulm, Germany). Textural properties were characterized by N_2_-sorption isotherms measured at 77 K (Nanoporosity-XQ, Mirae Scientific Instruments, Anyang, Korea). Elemental composition of the prepared catalysts (Fe ratio) was determined by inductively coupled plasma optical emission spectroscopy (ICP-OES, 700-ES, Varian, CA, USA).

### 2.3. Electrochemical Characterization

Electrochemical characteristics were measured in a conventional three electrode cell with N_2_ or O_2_ saturated 0.1 M KOH solution, using a potentiostat (Ivium technologies, EIN, The Netherlands) equipped with a rotating disk electrode (Pine research, Durham, NC, USA). Ag/AgCl (3 M NaCl) electrodes and Pt wire were used as reference and counter electrodes, respectively. All potentials were referred to the reversible hydrogen electrode (RHE) without specification. Working electrodes were prepared by dispersing 20 mg catalyst in 2 mL water/ethanol solvent (1:1 *v/v*) and 40 µL 5% Nafion solution, and then 20 µL catalyst slurry was pipetted onto a glassy carbon electrode (0.19635 cm^2^). Linear sweep voltammetry (LSV) measurements were performed at 5 mV/s scan rate of 1600 rpm, measured after 20 cyclic voltammetry tests from 0 to 1.2 V to stabilize the current. Durability was investigated by subjecting the samples to 6000 cycles of repeated potential ramp from 1.2 to 0 V.

## 3. Results and Discussion

### 3.1. Preparation and Physical Chracterizaton of FeS Catalysts

Scheme 1 shows the schematic fabrication procedure for bare and carbon-supported FeS catalysts. For FeS/N,S:CNT–GR, iron precursor reacts with ethanol to form Fe-ethoxide and was then mixed with CNT–GO in ethanol solution. Fe-thiourea complex was generated on the CNT–GO support by adding thiourea, and a powder product was obtained by annealing the Fe-thiourea complex/CNT–GO at 973 K for 3 h under flowing N_2_. The annealing produced mixed FeS, Fe_x_C, and Fe_x_N crystalline phases (Appendix A), but Fe_x_C and Fe_x_N phases disappeared after the acid treatment, leaving pure FeS phase due to better chemical stability of FeS under acid solution than Fe_x_C and Fe_x_N. Crystallization of FeS and reduction of GO to GR proceeded simultaneously during annealing. The N and S-doping into CNT–GR supports was also achieved using thiourea as a source of N and S. This synthetic procedure produced FeS nanoparticles with an average size of 24 nm dispersed on CNT–GR supports. Other carbon-supported FeS catalysts, FeS/N,S:CNT, and FeS/N,S:GR, were prepared following the same synthetic method employed with either CNT or GO, exclusively. Bare FeS was also similarly prepared without carbon support. The CNT–GR hybrid support can provide a large surface area for enhanced contact between FeS nanoparticles and electrolyte [12,29].

Single carbon support CNT or GO tends to bundle or stack together by itself, significantly limiting the carbon surface to form active sites and thus lowering electrocatalytic activity [30,31,32]. In contrast, the CNT–GR composite support created a three-dimensional open structure, avoiding bundling and stacking [29]. The CNT–GR composite also provides a good electron conducting pathway for the FeS nanoparticles. N and S-doping to carbon supports can enhance the ORR performance by redistributing spin and charge densities [25,26]. Thus, N,C:CNT–GR could be an effective catalyst support to enhance ORR activity, combining high conductivity and large surface area.

Figure 1 shows typical TEM images for prepared catalysts. The TEM image of bare FeS in Figure 1a shows a lattice spacing of 0.299 nm corresponding to the FeS (100) plane. Bare FeS particles were aggregated forming large clusters with approximately 700 nm diameter. In contrast, substantially reduced particle aggregation occurs for carbon-supported FeS catalysts, as shown in Figure 1b–d, with much smaller FeS nanoparticles (20–30 nm) distributed on each carbon support. Metal precursors are attracted by oxygen-containing functional groups within the carbon supports (CNT and GO). Hence FeS particles grow selectively on carbon supports (CNT, GR, and CNT–GR). Strong coupling between FeS particles and carbon supports mitigates FeS nanoparticle aggregation, e.g., Figure 1b shows FeS nanoparticles (28 nm) anchored on CNT without severe aggregation. No free-standing particles occurred, indicating that the CNT support mediated FeS growth and suppressed particle aggregation. Figure 1c shows GR layers with a wrinkled paper-like morphology and FeS nanoparticles (36 nm) distributed on the GR layers. Figure 1d shows a mixed CNT and GR morphology with FeS nanoparticles (24 nm) distributed on the CNT–GR composite supports. The inset of Figure 1d shows a lattice spacing of 0.265 nm, corresponding to the FeS (101) plane. The EDS mapping images of FeS/C,N:CNT–GR (Appendix A) show that Fe, S, and N overlap with C, suggesting FeS nanoparticles are dispersed on the N, S-doped CNT–GR support.

Figure 2a shows XRD patterns for prepared catalysts. Common peaks at 30°, 34°, 43°, and 52° can be indexed to hexagonal FeS (JCPDS 03-065-3408). Carbon-supported FeS samples show broad peaks at 26°, originating from CNT or GR. As mentioned, impurity peaks such as Fe_x_C or Fe_x_N were not present after acid treatment.

Figure 2b shows Raman spectra from the prepared catalysts. Intense peaks occur at 1350 (D) and 1580 cm^−1^ (G) and the numbers indicate their intensity ratios, i.e., I_D_/I_G_ ratios [33]. The D peak is related to sp^2^ ring disorder or defects, and the G peak to first order scattering from sp^2^ domains E_2g_ mode. I_D_/I_G_ measures the degree of disorder, where increased I_D_/I_G_ implies sp^2^ carbon restoration and smaller sp^2^ domains due to GO reduction [34]. Thus, the increased I_D_/I_G_ ratio in the FeS/GR (1.191) and the FeS/CNT–GR (1.190) compared with of GO (0.937) verifies thermal reduction of GO to GR during the synthesis.

Chemical states of FeS/CNT–GR were investigated by X-ray photoelectron spectroscopy (XPS). Figure 2c shows high resolution Fe 2p XPS spectra for FeS/CNT–GR. The peaks centered around 710.1 and 723.5 eV are due to Fe^2+^ 2p_3/2_ and Fe^2+^ 2p_1/2_ of FeS, respectively. The peaks at 713.3 eV (Fe^3+^ 2p_3/2_) and 727.3 eV (Fe^3+^ 2p_1/2_) suggest partial oxidation of the catalyst surface. [35,36]. Figure 2d shows N 1s spectra, with peaks at 398.5, 399.6, 401.1, and 403.8 eV corresponding to pyridinic, pyrrolic, graphitic, and oxidized N species, respectively [37]. Surface nitrogen content due to N-doping was 6.0 at% from the XPS survey scan. N-doping to carbon improves ORR performance by enhancing electrical conductivity of carbon or increasing defect sites of ORR activity [38,39]. Appendix A shows high resolution S 2p XPS spectra for FeS/N,S:CNT–GR, which could be deconvoluted into five peaks. Peaks at 161.2 and 162.8eV are related to S^2−^ 2p_3/2_ and S^2−^ 2p_1/2_ in FeS, respectively; Peaks at 164.5 and 165.5 eV originate from polysulfide S in the carbon plane; the peak at 168.2 eV is attributed to sulphate (SO_4_^2−^) species due to acid treatment or partial oxidation of sulfide upon air exposure [40,41]. In S 2p spectra of N, S-doped CNT–GR made without Fe precursor (Appendix A), FeS-related peaks disappeared while the peaks attributed to polysulfide S in the carbon plane and sulphate species were maintained. The results indicate that the thiourea was the sulfur source for FeS crystallization and S-doping to the carbon supports. Appendix A shows XPS spectra for FeS/N,S:CNT and FeS/N,S:GR. They exhibit similar Fe 2p, N 1s, and S 2p peaks to FeS/N,S:CNT–GR, suggesting similar chemical states of FeS for all the carbon-supported catalysts.

Textural properties for the prepared catalysts were investigated by N_2_ adsorption-desorption isotherms, and compared with other TMS/carbon catalysts in Appendix A. Figure 3a shows that carbon-supported FeS samples exhibited the type IV isotherm with a typical hysteresis of the isotherms, indicating the presence of mesopores, whereas bare FeS does not show clear hysteresis. Brunauer–Emmett–Teller (BET) surface area for bare FeS was 14 m^2^g^−1^, whereas FeS/N,S:CNT–GR achieved significantly improved BET surface area of 191 m^2^g^−1^ after introducing the N,S:CNT–GR support. FeS/N,S:CNT and FeS/N,S:GR also showed increased BET surface areas of 174 and 137 m^2^g^−1^, respectively. The larger surface area of FeS/N,S:CNT–GR compared with FeS/N,S:CNT and FeS/N,S:GR was due to a synergy between CNT and GR acting as spacers for each other, alleviating CNT’s bundling and GR layers’ stacking [12]. Pore size distribution was determined using the desorption isotherm following the Barrett–Joyner–Halenda method. Average pore size for all carbon-supported FeS catalysts was ~4 nm (Figure 3b), and pore volume varied as FeS/N,S:CNT–GR (0.5568 cm^3^/g) > FeS/N,S:CNT (0.5067 cm^3^/g) > FeS/N,S:GR (0.4077 cm^3^/g). The large surface area with abundant mesopores will stabilize smaller FeS nanoparticles, leading to improved catalytic activity for ORR [42].

### 3.2. Electrochmical Chracterizaton of FeS Catalysts

Figure 4a shows the LSV results for prepared catalysts in O_2_-saturated 0.1M KOH solution. Benchmark commercial 20% Pt/C (Johnson–Matthey) achieved the best ORR performance with higher onset potential E_onset_ = 1.0 V and half-wave potential E_1/2_ = 0.84 V than others. The FeS/N,S:CNT–GR catalyst showed comparable performance to Pt/C, but outperformed the other FeS catalysts in ORR activity. Thus, Eonset values were 0.923, 0.931, and 0.972 V for FeS/N,S:GR, FeS/N,S:CNT, and FeS/N,S:CNT–GR, respectively. The E_1/2_ value for FeS/N,S:CNT–GR (0.827 V) was higher than FeS/N,S:CNT (0.808 V) and FeS/N,S:GR (0.800 V). Bare FeS showed poor ORR activity with an Eonset of 0.8 V and E_1/2_ of 0.58 V. Thus, carbon supports enhanced ORR activity substantially, demonstrating their critical importance, providing high conductivity and large surface area for the loaded FeS nanoparticles. Besides, dual N, S-doping to the carbon supports further enhanced the activity due to changing the charge distribution in the carbon framework and improving electrical conductivity compared with undoped carbon supports [25,26,43,44]. These effects were maximized in FeS/N,S:CNT–GR, achieving a top performance compared to previously reported iron or other TMS-based catalysts (Appendix A).

For further investigation of the ORR pathway and kinetics, LSVs of FeS/CNT–GR at different rotation speeds were measured and the corresponding Koutecky–Levich (K–L) plots were obtained at the potential range of 0.2–0.7 V. Figure 4b shows that the current density of FeS/N,S:CNT–GR increases with increasing rotation speed at the same potential due to enhanced oxygen diffusion on the electrode. Appendix A shows Koutecky–Levich (K–L) plots for FeS/N,S:CNT–GR described as
(1)$$\frac{1}{J}=\frac{1}{J_{D}}+\frac{1}{J_{K}}=\frac{1}{{B\omega }^{1/2}}+\frac{1}{J_{K}}$$
where J is measured current density; J_D_ is diffusion-limited current density; J_K_ is kinetic current density; ω is electrode rotation speed;
B = 0.2nFC_0_D^2/3^υ^−1/6^(2)
is the K–L plot slope, where n is electron transfer number, F is Faraday’s constant (96,486 C mol^−1^), C_0_ is bulk concentration of oxygen in 0.1M KOH solution (1.2 × 10^−6^ mol cm^−3^), D is diffusion coefficient of oxygen in 0.1M KOH (1.9 × 10^−5^ cm^2^·s^−1^), and υ is kinetic viscosity of oxygen in 0.1M KOH (1.0 × 10^−2^ cm^2^ s^−1^) [45]. The obtained electron transfer numbers are close to 4.0 (Appendix A), indicating the dominant four electron ORR catalytic pathway proceeds for the FeS/N,S:CNT–GR catalyst.

Figure 4c shows LSV curves for FeS/N,S:CNT–GR after 6000 potential cycles between 1.2 and 0 V. Activity loss for FeS/N,S:CNT–GR was marginal with slightly decreased E_1/2_ = 0.821 V from 0.827 V, whereas commercial Pt/C recorded a 30 mV decrease in E_1/2_. Similarly, the current loss for the FeS/N,S:CNT–GR was ~6% after continuous operation for 10,000 s at 0.7 V (Figure 4d), whereas commercial Pt/C showed much faster current decay of 27%, indicating that our FeS/N,S:CNT has a high ORR activity comparable to Pt/C, and much better ORR stability than Pt/C.

This excellent ORR activity and stability of the FeS/N,S:CNT–GR catalyst could arise from synergy between small FeS nanoparticles and the N, S dual-doped CNT–GR support. The N,S:CNT–GR support provides a high electrical conductivity and a large surface area, improving FeS activity and electrolyte contact to active sites (FeS). Indeed, in Appendix A, FeS/N,S:CNT–GR achieved considerably higher double layer capacitance C_dl_ of 36.6 mF/cm^2^ compared with FeS/N,S:GR, FeS/N,S:CNT, and bare FeS (C_dl_ values of 10.5, 24.6, and 1.16 mF/cm^2^, respectively) [46]. The C_dl_ value is proportional to contact area between electrode and electrolyte (or electrochemical surface area, ECSA), hence the N,S:CNT–GR support efficiently increases the contact area, alleviating CNT bundling and GR layers stacking [12,39]. Furthermore, simultaneous N and S-doping to CNT–GR was easily achieved by employing thiourea, improving the ORR performance by redistributing spin and charge densities [21,24]. The CNT–GR support also mediates FeS growth, reducing particle aggregation and further increasing FeS reaction sites. Appendix A shows that the N, S dual-doped CNT–GR prepared by identical synthetic methods without Fe precursor has its own ORR activity, achieving E_1/2_ of 0.762 V. Hence, combining it with FeS considerably improved ORR activity of the FeS/N,S:CNT–GR catalyst, indicating a synergy between FeS and N,S:CNT–GR. Thus, considering the facile synthetic method and high ORR performance, FeS/N,S:CNT–GR catalysts constitute a potential candidate to substitute commercial Pt/C catalyst.

## 4. Conclusions

This work successfully prepared a non-precious metal catalyst for ORR comprising FeS nanoparticles dispersed onto N, S dual-doped CNT–GR composite supports through a simple annealing and acid treatment, achieving simultaneous FeS formation and N, S dual-doping into CNT–GR. The synthesized FeS/N,S:CNT–GR catalyst exhibited the highest ORR performance among prepared FeS-based catalysts with a small E_1/2_ value of 0.827 V, comparable to commercial Pt/C. Improved ORR performance was attributed to a synergy between the small FeS nanoparticles with high activity and N, S dual-doped CNT–GR support providing improved high electrical conductivity, large surface area, and its own ORR performance caused by modified electronic structure by the dual doping. Thus, N,S:CNT–GR composite support mediates FeS growth, reducing particle aggregation, and further increasing reaction sites. This FeS/N,S:CNT–GR catalyst offers a potential non-precious metal ORR catalyst of a high activity with a good stability.

## Data Availability

All data is contained within the article.

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
