# Peer review of "Nanostructured Iron Sulfide/N, S Dual-Doped Carbon Nanotube-Graphene Composites as Efficient Electrocatalysts for Oxygen Reduction Reaction"

_materials, 2021, doi:10.3390/ma14092146_

Round 1

Reviewer 1 Report

The authors synthesized the FeS/N,S:CNT-GR nanostructure to investigate the catalytic behavior for for oxygen reduction reaction. They also compared the structure with FeS, FeS/N,S:GR, and FeS/N,S:CNT. TEM, XRD, Raman, and XPS are employed to do material characterizations. The results from LSV, K-L plot, and stability test suggest that the FeS/N,S:CNT-GR nanostructure has better ORR performance than the other samples and commercial Pt/C catalyst. The results are significant and well discussed. The manuscript may be published in Materials after minor revisions.

1. Line 36: The abbreviation "TMS" should be defined. What is "TM" abbreviated for? 

2. Line 107: The FexC, and FexN crystalline phases in Figure S1 should be marked.

3. Some numbers need to be corrected.
Line 128: 2.99 nm => 0.299 nm
Line 162: 727.3 eV (Fe3+ 2p3/2) => 2p1/2
Line 163: 396.6 => I think 399.6 

Reviewer 2 Report

The manuscript “Nanostructured Iron Sulfide/N, S Dual-doped Carbon Nanotube-Graphene Composites as Efficient Electrocatalysts for Oxygen Reduction Reaction” deals with the production of a new catalyst, at enhanced electrochemical performance. The work is ambitious and well organized. Intriguing results have been obtained. Therefore, the publication is recommended; but, some revisions are required.

Detailed comments:

- Introduction. The state of the art can be enlarged, adding some works in the field, related to the improvement of the electrochemical performance of graphene oxide when loaded in aerogels. Please, see the work of Sarno et al., SC-CO2-assisted process for a high energy density aerogel supercapacitor: The effect of GO loading, Nanotechnology, 2017, 28, Article number 204001; etc..

Also the novelty of this research work should be better highlighted in the last part of the Introduction.

- Results. Morphological results of the catalysts, i.e., N2 adsorption-desorption isotherms, BET surface area, pore size, should be compared and discussed with respect to the ones present in the literature.

Round 2

Reviewer 2 Report

The authors performed the modifications proposed by the Reviewer and improved the manuscript.